# Generalizing GANs: A Turing Perspective

**Roderich Groß and Yue Gu**
Department of Automatic Control and Systems Engineering
The University of Sheffield
`{r.gross,ygu16}@sheffield.ac.uk`

**Wei Li**
Department of Electronics
The University of York
`wei.li@york.ac.uk`

**Melvin Gauci**
Wyss Institute for Biologically Inspired Engineering
Harvard University
`mgauci@g.harvard.edu`

## Abstract

Recently, a new class of machine learning algorithms has emerged, where models and discriminators are generated in a competitive setting. The most prominent example is Generative Adversarial Networks (GANs). In this paper we examine how these algorithms relate to the Turing test, and derive what—from a Turing perspective—can be considered their defining features. Based on these features, we outline directions for generalizing GANs—resulting in the family of algorithms referred to as *Turing Learning*. One such direction is to allow the discriminators to interact with the processes from which the data samples are obtained, making them "interrogators", as in the Turing test. We validate this idea using two case studies. In the first case study, a computer infers the behavior of an agent while controlling its environment. In the second case study, a robot infers its own sensor configuration while controlling its movements. The results confirm that by allowing discriminators to interrogate, the accuracy of models is improved.

## 1   Introduction

Generative Adversarial Networks [1] (GANs) are a framework for inferring generative models from training data. They place two neural networks—a model and a discriminator—in a competitive setting. The discriminator's objective is to correctly label samples from either the model or the training data. The model's objective is to deceive the discriminator, in other words, to produce samples that are categorized as training data by the discriminator. The networks are trained using a gradient-based optimization algorithm. Since their inception in 2014, GANs have been applied in a range of contexts [2, 3], but most prominently for the generation of photo-realistic images [1, 4].

In this paper we analyze the striking similarities between GANs and the Turing test [5]. The Turing test probes a machine's ability to display behavior that, to an interrogator, is indistinguishable from that of a human. Developing machines that pass the Turing test could be considered as a canonical problem in computer science [6]. More generally, the problem is that of imitating (and hence inferring) the structure and/or behavior of any system, such as an organism, a device, a computer program, or a process.

The idea to infer models in a competitive setting (model versus discriminator) was first proposed in [7]. The paper considered the problem of inferring the behavior of an agent in a simple environment. The behavior was deterministic, simplifying the identification task. In a subsequent work [8], the method, named *Turing Learning*, was used to infer the behavioral rules of a swarm of memoryless

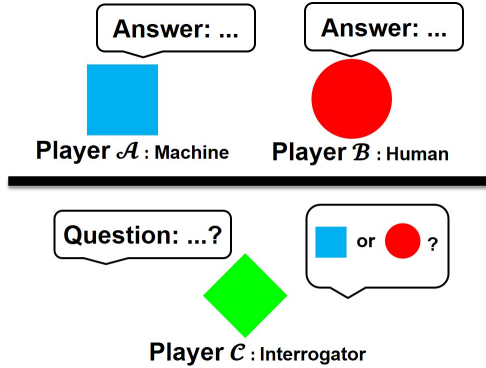

Figure 1: Illustration of the Turing test setup introduced in [5]. Player $\mathcal{C}$ (the interrogator) poses questions to and receives labelled answers from players $\mathcal{A}$ and $\mathcal{B}$. Player $\mathcal{C}$ does not know which label (blue square or red disk) corresponds to which player. Player $\mathcal{C}$ has to determine this after questioning.

robots. The robot's movements were tracked using an external camera system, providing the training data. Additional robots executed the rules defined by the models.

The contributions of this paper are

- to examine the defining features of GANs (and variants)—assuming a Turing perspective;
- to outline directions for generalizing GANs, in particular, to encourage alternative implementations and novel applications; for example, ones involving physical systems;
- to show, using two case studies, that more accurate models can be obtained if the discriminators are allowed to *interact* with the processes from which data samples are obtained (as the interrogators in the Turing test).[1]

## 2 A Turing Perspective

In 1950, Turing proposed an imitation game [5] consisting of three players $\mathcal{A}$, $\mathcal{B}$ and $\mathcal{C}$. Figure 1 shows a schematic of this game. Player $\mathcal{C}$, also referred to as the interrogator, is unable to see the other players. However, the interrogator can pose questions to and receive answers from them. Answers from the same player are consistently labelled (but not revealing its identity, $\mathcal{A}$ or $\mathcal{B}$). At the end of the game, the interrogator has to guess which label belongs to which player. There are two variants of the game, and we focus on the one where player $\mathcal{A}$ is a machine, while player $\mathcal{B}$ is human (the interrogator is always human). This variant, depicted in Figure 1, is commonly referred to as the Turing test [9, 10]. To pass the test, the machine would have to produce answers that the interrogator believes to originate from a human. If a machine passed this test, it would be considered intelligent.

For GANs (and variants), player $\mathcal{C}$, the interrogator, is no longer human, but rather a computer program that learns to discriminate between information originating from players $\mathcal{A}$ and $\mathcal{B}$. Player $\mathcal{A}$ is a computer program that learns to trick the interrogator. Player $\mathcal{B}$ could be any system one wishes to imitate, including humans.

### 2.1 Defining Features of GANs

Assuming a Turing perspective, we consider the following as the defining features of GANs (and variants):

- a training agent, $\mathcal{T}$, providing genuine data samples (the training data);
- a model agent, $\mathcal{M}$, providing counterfeit data samples;

- a discriminator agent, $\mathcal{D}$, labelling data samples as either genuine or counterfeit;
- a process by which $\mathcal{D}$ observes or interacts with $\mathcal{M}$ and $\mathcal{T}$;
- $\mathcal{D}$ and $\mathcal{M}$ are being optimized:
    - $\mathcal{D}$ is rewarded for labelling data samples of $\mathcal{T}$ as genuine;
    - $\mathcal{D}$ is rewarded for labelling data samples of $\mathcal{M}$ as counterfeit;
    - $\mathcal{M}$ is rewarded for misleading $\mathcal{D}$ (to label its data samples as genuine).

It should be noted that in the Turing test there is a bi-directional exchange of information between player $\mathcal{C}$ and either player $\mathcal{A}$ or $\mathcal{B}$. In GANs, however, during any particular "game", data flows only in one direction: The discriminator agent receives data samples, but is unable to influence the agent at the origin during the sampling process. In the case studies presented in this paper, this limitation is overcome, and it is shown that this can lead to improved model accuracy. This, of course, does not imply that active discriminators are beneficial for every problem domain.

## 2.2 Implementation Options of (Generalized) GANs

GANs and their generalizations, that is, algorithms that possess the aforementioned defining features, are instances of *Turing Learning* [8]. The *Turing Learning* formulation removes (from a Turing perspective unnecessary) restrictions of the original GAN formulation, for example, the need for models and discriminators to be represented as neural networks, or the need for optimizing these networks using gradient descent. As a result of this, the *Turing Learning* formulation is very general, and applicable to a wide range of problems (e.g., using models with discrete, continuous or mixed representations).

In the following, we present the aspects of implementations that are not considered as defining features, but rather as implementation options. They allow *Turing Learning* to be tailored, for example, by using the most suitable model representation and optimization algorithm for the given problem domain. Moreover, users can choose implementation options they are familiar with, making the overall framework[2] more accessible.

- *Training data*. The training data could take any form. It could be artificial (e.g., audio, visual, textual data in a computer), or physical (e.g., a geological sample, engine, painting or human being).

- *Model presentation*. The model could take any form. In GANs [1], it takes the form of a neural network that generates data when provided with a random input. Other representations include vectors, graphs, and computer programs. In any case, the representation should be expressive enough, allowing a model to produce data with the same distribution as the training data. The associated process could involve physical objects (e.g., robots [8]). If the training data originates from physical objects, but the model data originates from simulation, special attention is needed to avoid the so called reality gap [11]. Any difference caused not by the model but rather the process to collect the data (e.g., tracking equipment) may be detected by the discriminators, which could render model inference impossible.

- *Discriminator representation*. The discriminator could take any form. Its representation should be expressive enough to distinguish between genuine and counterfeit data samples. These samples could be artificial or physical. For example, a discriminator could be networked to an experimental platform, observing and manipulating some physical objects or organisms.

- *Optimization algorithms*. The optimization algorithms could take any form as long as they are compatible with the solution representations. They could use a single candidate solution or a population of candidate solutions [8, 12]. In the context of GANs, gradient-based optimization algorithms are widely applied [13]. These algorithms however require the objective function to be differentiable and (ideally) unimodal. A wide range of metaheuristic algorithms [14] could be explored for domains with more complex objective functions. For example, if the model was represented using a computer program, genetic programming algorithms could be used.

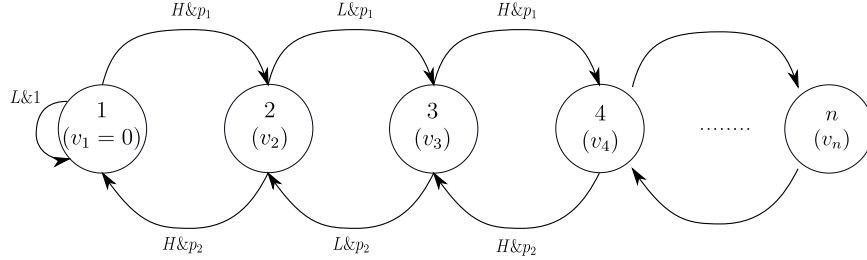

Figure 2: In Case Study 1, we consider a non-embodied agent that is subjected to a stimulus, $S$, which can be either low ($L$) or high ($H$). The task is to infer how the agent responds to the stimulus. The discriminator controls the stimulus while observing the behavior of the agent (expressed as $v$), which is governed by above probabilistic finite-state machine. Label $S\&p$ denotes that if the stimulus is $S \in \{L, H\}$, the corresponding transition occurs with probability $p$. We assume that the structure of the state machine is known, and that the parameters $(p_1, p_2, v_2, v_3, \ldots, v_n)$ are to be inferred.

- *Coupling mechanism between the model and discriminator optimizers*. The optimization processes for the model and discriminator solutions are dependent on each other. Hence they may require careful synchronization [1]. Moreover, if using multiple models and/or multiple discriminators, choices have to be made for which pairs of solutions to evaluate. Elaborate evaluation schemes may take into account the performance of the opponents in other evaluations (e.g., using niching techniques). Synchronization challenges include those reported for coevolutionary systems.[3] In particular, due to the so-called *Red Queen Effect*, the absolute quality of solutions in a population may increase while the quality of solutions relative to the other population may decrease, or vice versa [18]. *Cycling* [20] refers to the phenomenon that some solutions that have been lost, may get rediscovered in later generations. A method for overcoming the problem is to retain promising solutions in an archive—the "hall of fame" [21]. *Disengagement* can occur when one population (e.g., the discriminators) outperforms the other population, making it hard to reveal differences among the solutions. Methods for addressing disengagement include "resource sharing" [22] and "reducing virulence" [20].

- *Termination criterion*. Identifying a suitable criterion for terminating the optimization process can be challenging, as the performance is defined in relative rather than absolute terms. For example, a model that is found to produce genuine data by each of a population of discriminators may still not be useful (the discriminators may have performed poorly). In principle, however, any criterion can be applied (e.g., convergence data, fixed time limit, etc).

## 3   Case Study 1: Inferring Stochastic Behavioral Processes Through Interaction

### 3.1   Problem Formulation

This case study is inspired from ethology—the study of animal behavior. Animals are sophisticated agents, whose actions depend on both their internal state and the stimuli present in their environment. Additionally, their behavior can have a stochastic component. In the following, we show how *Turing Learning* can infer the behavior of a simple agent that captures the aforementioned properties.

The agent's behavior is governed by the probabilistic finite-state machine (PFSM)[4] shown in Figure 2. It has $n$ states, and it is assumed that each state leads to some observable behavioral feature, $v \in \mathbb{R}$, hereafter referred to as the agent's velocity. The agent responds to a stimulus that can take two levels, low ($L$) or high ($H$). The agent starts in state 1. If the stimulus is $L$, it remains in state 1 with certainty.

If the stimulus is $H$, it transitions to state 2 with probability $p_1$, and remains in state 1 otherwise. In other words, on average, it transitions to state 2 after $1/p_1$ steps. In state $k = 2, 3, \ldots, n - 1$, the behavior is as follows. If the stimulus is identical to that which brings the agent into state $k$ from state $k - 1$, the state reverts to $k - 1$ with probability $p_2$ and remains at $k$ otherwise. If the stimulus is different to that which brings the agent into state $k$ from state $k - 1$, the state progresses to $k + 1$ with probability $p_1$ and remains at $k$ otherwise. In state $n$, the only difference is that if the stimulus is different to that which brought about state $n$, the agent remains in state $n$ with certainty (as there is no next state to progress to).

By choosing $p_1$ close to 0 and $p_2 = 1$, we force the need for interaction if the higher states are to be observed for a meaningful amount of time. This is because once a transition to a higher state happens, the interrogator must immediately toggle the stimulus to prevent the agent from regressing back to the lower state.

### 3.2 *Turing Learning* Implementation

We implement *Turing Learning* for this problem as follows:

- *Training data.* To obtain the training data, the discriminator interacts with the PFSM, shown in Figure 2. The number of states are set to four ($n = 4$). The parameters used to generate the (genuine) data samples are given by:

$$\mathbf{q} = (p_1^*, p_2^*, v_2^*, v_3^*, v_4^*) = (0.1, 1.0, 0.2, 0.4, 0.6). \tag{1}$$

- *Model representation.* It is assumed that the structure of the PFSM is known, while the parameters, $\mathbf{q}$, are to be inferred. All parameters can vary in $\mathbb{R}$. To interpret $p_1$ and $p_2$ as probabilities, they are mapped to the closest point in $[0, 1]$, if outside this interval. The model data is derived analogously to that of the training data.

- *Discriminator representation.* The discriminator is implemented as an Elman neural network [25] with 1 input neuron, 5 hidden neurons, and 2 output neurons. At each time step $t$, the observable feature (the agent's velocity $v$) is fed into the input neuron.[5] After updating the neural network, the output from one of the output neurons is used to determine the stimulus at time step $t + 1$, $L$ or $H$. At the end of a trial (100 time steps), the output from the other output neuron is used to determine whether the discriminator believes the agent under investigation to be the training agent ($\mathcal{T}$) or model agent ($\mathcal{M}$).

- *Optimization Algorithms.* We use a standard $(\mu + \lambda)$ evolution strategy with self-adapting mutation strengths [26] for both the model and the discriminator populations. We use $\mu = \lambda = 50$ in both cases. The populations are initialized at random. The parameter values of the optimization algorithm are set as described in [26].

- *Coupling mechanism between the model and discriminator optimizers.* The coupling comes from the evaluation process, which in turn affects the population selection. Each of the 100 candidate discriminators is evaluated once with each of the 100 models, as well as an additional 100 times with the training agent. It receives a point every time it correctly labels the data as either genuine or counterfeit. At the same time, each model receives a point for each time a discriminator mistakenly judges its data as genuine.

- *Termination criterion.* The optimization process is stopped after 1000 generations.

### 3.3 Results

To validate the advantages of the interactive approach, we use three setups for the *Turing Learning* algorithm. In the default setup, hereafter "Interactive" setup, the discriminator controls the environmental stimulus while observing the agent. In the other two setups, the discriminator observes the agent in a passive manner; that is, its output is not used to update the stimulus. Instead, the stimulus is uniformly randomly chosen at the beginning of the trial, and it is toggled with probability 0.1 at any time step (the stimulus is hence expected to change on average every 10 time steps). In setup "Passive 1", the discriminator has the same input as in the "Interactive" setup (the observable feature, $v$). In setup "Passive 2", the discriminator has one additional input, the current stimulus ($S$). All other aspects of the passive setups are identical to the "Interactive" setup.

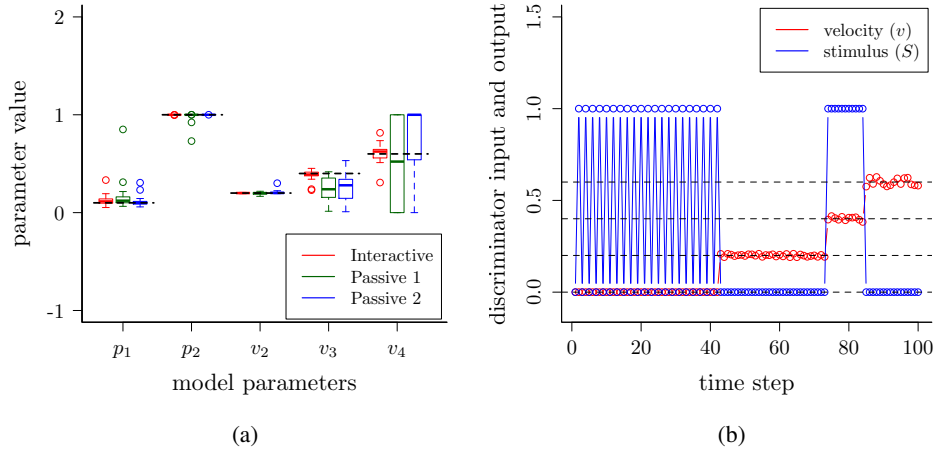

(a)                                           (b)

Figure 3: Results from Case Study 1. (a) Model parameters *Turing Learning* inferred about a simulated agent. In the "Interactive" setup, the discriminator observes the agent while controlling a stimulus that the agent responds to. In the two passive setups, the discriminator observes the agent and/or stimulus, while the latter is randomly generated (for details, see text). The models are those with the highest evaluation value in the final generation (20 runs per setup). The dashed lines indicate the optimal parameter value (which is to be identified). (b) Example showing how one of the discriminators interacted with the agent during a trial. For the stimulus (blue), L and H are shown as 0 and 1, respectively.

For each setup, we performed 20 runs of the *Turing Learning* algorithm. Figure 3(a) shows the distribution of the inferred models that achieved the highest evaluation value in the 1000th generation. The "Interactive" setup is the only one that inferred all parameters with good accuracy.

Figure 3(b) shows a typical example of how a discriminator interacts with the agent. The discriminator initially sets the environmental stimulus to alternating values (i.e., toggling between $H$ and $L$). Once the agent advances from state 1 to state 2, the discriminator instantly changes the stimulus to $L$ and holds it constant. Once the agent advances to higher states, the stimulus is switched again, and so forth. This strategy allows the discriminator to observe the agent's velocity in each state.

# 4 Case Study 2: A Robot Inferring Its Own Sensor Configuration

## 4.1 Problem Formulation

The reality gap is a well-known problem in robotics: Often, behaviors that work well in simulation do not translate effectively into real-world implementations [11]. This is because simulations are generally unable to capture the full range of features of the real world, and therefore make simplifying assumptions. Yet, simulations can be important, even on-board a physical robot, as they facilitate planning and optimization.

This case study investigates how a robot can use *Turing Learning* to improve the accuracy of a simulation model of itself, though a process of self-discovery, similar to [27]. In a practical scenario, the inference could take place on-board a physical platform. For convenience, we use an existing simulation platform [28], which has been extensively verified and shown to be able to cross the reality gap [29]. The robot, an e-puck [30], is represented as a cylinder of diameter 7.4 cm, height 4.7 cm and mass 152 g. It has two symmetrically aligned wheels. Their ground contact velocity ($v_{\text{left}}$ and $v_{\text{right}}$) can be set within [-12.8, 12.8] (cm/s). During the motion, random noise is applied to each wheel velocity, by multiplying it with a number chosen with a uniform distribution in the range (0.95, 1.05).

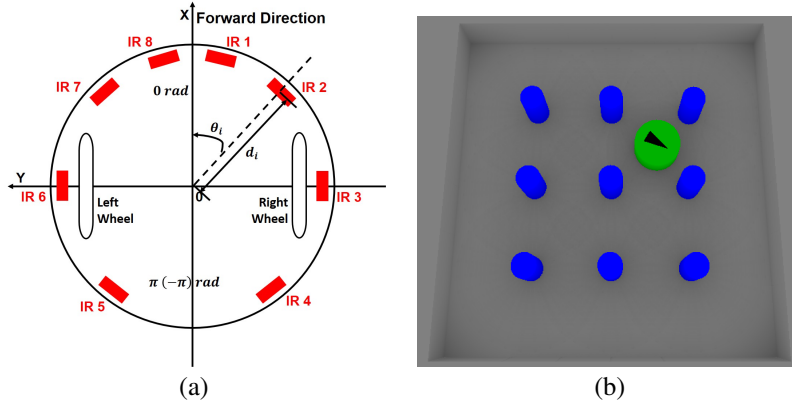

(a)                                                      (b)

Figure 4: In Case Study 2, we consider a miniature mobile robot, the e-puck, that perceives its environment via eight infrared (IR) proximity sensors. The robot is unaware of the spatial configuration of these sensors, and has to infer it. The discriminator controls the movements of the robot, while observing the reading values of the sensors. (a) The sensor configuration to be inferred is the one of the physical e-puck robot. It comprises of 16 parameters, representing the orientations ($\boldsymbol{\theta}$) and displacements ($\mathbf{d}$) of the 8 proximity sensors. (b) The robot is placed at random into an environment with nine moveable obstacles.

The robot has eight infrared proximity sensors distributed around its cylindrical body, see Figure 4(a). The sensors provide noisy reading values $(s_1, s_2, \ldots, s_8)$. We assume that the robot does not know where the sensors are located (neither their orientations, nor their displacements from the center). Situations like this are common in robotics, where uncertainties are introduced when sensors get mounted manually or when the sensor configuration may change during operation (e.g., at the time of collision with an object, or when the robot itself reconfigures the sensors). The sensor configuration can be described as follows:

$$\mathbf{q} = (\theta_1, \theta_2, \ldots, \theta_8, d_1, d_2, \ldots, d_8), \tag{2}$$

where $d_i \in (0, R]$ defines the distance of sensor $i$ from the robot's center ($R$ is the robot's radius), and $\theta_i \in [-\pi, \pi]$ defines the bearing of sensor $i$ relative to the robot's front.

The robot operates in a bounded square environment with sides 50 cm, shown in Figure 4(b). The environment also contains nine movable, cylindrical obstacles, arranged in a grid. The distance between the obstacles is just wide enough for an e-puck to pass through.

## 4.2   *Turing Learning* Implementation

We implement *Turing Learning* for this problem as follows:

- *Training data*. The training data comes from the eight proximity sensors of a "real" e-puck robot, that is, using sensor configuration $\mathbf{q}$ as defined by the robot (see Figure 4(a)). The discriminator controls the movements of the robot within the environment shown in Figure 4(b), while observing the readings of its sensors.

- *Model representation*. It is assumed that the sensor configuration, $\mathbf{q}$, is to be inferred. In other words, a total of 16 parameters have to be estimated.

- *Discriminator representation*. As in Case Study 1, the discriminator is implemented as an Elman neural network with 5 hidden neurons. The network has 8 inputs that receive values from the robot's proximity sensors $(s_1, s_2, \ldots, s_8)$. In addition to the classification output, the discriminator has two control outputs, which are used to set the robot's wheel velocities ($v_{\text{left}}$ and $v_{\text{right}}$). In each trial, the robot starts from a random position and random orientation within the environment.[6] The evaluation lasts for 10 seconds. As the robot's sensors and actuators are updated 10 times per second, this results in 100 time steps.

- The remaining aspects are implemented exactly as in Case Study 1.

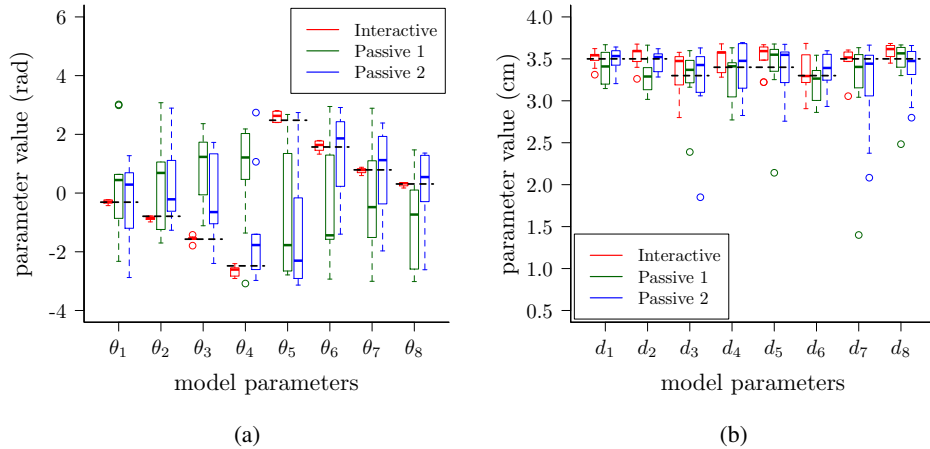

Figure 5: Results from Case Study 2. Model parameters *Turing Learning* inferred about the sensor configuration of the e-puck robot: (a) sensor orientations, (b) sensor displacements. In the "Interactive" setup, the discriminator observes the sensor reading values while controlling the movements of the robot. In the two passive setups, the discriminator observes the sensor reading values and/or movements while the latter are randomly generated (for details, see text). The models are those with the highest evaluation value in the final generation (20 runs per setup). The dashed lines indicate the optimal parameter value (which is to be identified).

### 4.3 Results

To validate the advantages of the interactive approach, we use again three setups. In the "Interactive" setup the discriminator controls the movements of the robot while observing its sensor readings. In the other two setups, the discriminator observes the robot's sensor readings in a passive manner; that is, its output is not used to update the movements of the robot. Rather, the pair of wheel velocities is uniformly randomly chosen at the beginning of the trial, and, with probability 0.1 at any time step (the movement pattern hence is expected to change on average every 10 time steps). In setup "Passive 1", the discriminator has the same inputs as in the "Interactive" setup (the reading values of the robot's sensors, $s_1, s_2, \ldots, s_8$). In setup "Passive 2", the discriminator has two additional inputs, indicating the velocities of the left and right wheels ($v_{\text{left}}$ and $v_{\text{right}}$). All other aspects of the passive setups are identical to the "Interactive" setup.

For each setup, we performed 20 runs of the *Turing Learning* algorithm. Figure 5 shows the distribution of the inferred models that achieved the highest evaluation value in the 1000th generation. The "Interactive" setup is the only one that inferred the orientations of the proximity sensors with good accuracy. The displacement parameters were inferred with all setups, though none of them was able to provide accurate estimates.

Figure 6 shows a typical example of how a discriminator controls the robot. At the beginning, the robot rotates clockwise, registering an obstacle with sensors $s_7, s_6, \ldots, s_2$ (in that order). The robot then moves forward, and registers the obstacle with sensors $s_1$ and/or $s_8$, while pushing it. This confirms that $s_1$ and $s_8$ are indeed forward-facing. Once the robot has no longer any obstacle in its front, it repeats the process. To validate if the sensor-to-motor coupling was of any significance for the discrimination task, we recorded the movements of a robot controlled by the best discriminator of each of the 20 runs. The robot used either the genuine sensor configuration (50 trials) or the best model configuration of the corresponding run (50 trials). In these 2000 "closed-loop" experiments, the discriminator made correct judgments in 69.45% of the cases. We then repeated the 2000 trials, now ignoring the discriminator's control outputs, but rather using the movements recorded earlier. In these 2000 "open-loop" experiments, the discriminator made correct judgments in 58.60% of the cases—a significant drop, though still better than guessing (50%).

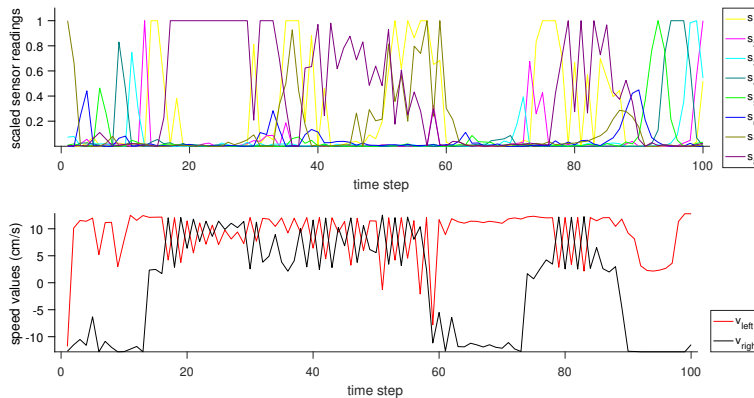

Figure 6: Example showing how one of the discriminators in Case Study 2 controlled the robot's movements during the trial. The discriminator takes as input the robot's eight sensor reading values (shown at the top), and controls the velocities of the wheels (shown at the bottom). The discriminator has to decide whether the sensor configuration of the robot corresponds to the one of the physical e-puck robot. For details, see text.

# 5   Conclusion

In this paper we analyzed how Generative Adversarial Networks (GANs) relate to the Turing test. We identified the defining features of GANs, if assuming a Turing perspective. Other features, including choice of model representation, discriminator representation, and optimization algorithm, were viewed as implementation options of a generalized version of GANs, also referred to as *Turing Learning*.

It was noted that the discriminator in GANs does not *directly* influence the sampling process, but rather is provided with a (static) data sample from either the generative model or training data set. This is in stark contrast to the Turing test, where the discriminator (the interrogator) plays an active role; it poses questions to the players, to reveal the information most relevant to the discrimination task. Such interactions are by no means always useful. For the purpose for generating photo-realistic images, for example, they may not be needed.[7] For the two case studies presented here, however, interactions were shown to cause an improvement in the accuracy of models.

The first case study showed how one can infer the behavior of an agent while controlling a stimulus present in its environment. It could serve as a template for studies of animal/human behavior, especially where some behavioral traits are revealed only through meaningful interactions. The inference task was not simple, as the agent's actions depended on a hidden stochastic process. The latter was influenced by the stimulus, which was set to either low or high by the discriminator (100 times). It was not known in advance which of the $2^{100}$ sequences are useful. The discriminator thus needed to dynamically construct a suitable sequence, taking the observation data into account.

The second case study focused on a different class of problems: active self-discovery. It showed that a robot can infer its own sensor configuration through controlled movements. This case study could serve as a template for modelling physical devices. The inference task was not simple, as the robot started from a random position in the environment, and its motors and sensors were affected by noise. The discriminator thus needed to dynamically construct a control sequence that let the robot approach an obstacle and perform movements for testing its sensor configuration.

Future work could attempt to build models of more complex behaviors, including those of humans.

**Acknowledgments**

The authors thank Nathan Lepora for stimulating discussions.

## Footnotes

[1]Different to [7], we consider substantially more complex case studies, where the discriminators are required to genuinely interact with the systems, as a pre-determined sequence of interventions would be unlikely to reveal all the observable behavioral features.

[2]For an algorithmic description of *Turing Learning*, see [8].

[3]Coevolutionary algorithms have been studied in a range of contexts [15, 16, 17], including system identification [18, 19], though these works differ from GANs and *Turing Learning* in that no discriminators evolve, but rather pre-defined metrics gauge on how similar the model and training data are. For some system identification problems, the use of such pre-defined metrics can result in poor model accuracy, as shown in [8].

[4]PFSMs generalize the concept of Markov chains [23, 24].

[5]To emulate a noisy tracking process, the actual speed value is multiplied with a number chosen with a uniform distribution in the range (0.95, 1.05).

[6]As the robot knows neither its relative position to the obstacles, nor its sensor configuration, the scenario can be considered as a *chicken-and-egg* problem.

[7]Though if the discriminator could request additional images by the same model or training agent, problems like mode collapse might be prevented.

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
