[Reviews · NeurIPS 2017]

Reviewer 1



GANs are a very interesting idea, which I haven't heard of so far. Using them for control is very appealing. The paper is very well written, easy to understand, high in quality: both in methods and results. There are just two comments. Co-evolution and resilience are much older than the authors assume. Co-evolution has been studied e.g. in Stefano Nolfi and Dario Floreano. 1998. Coevolving Predator and Prey Robots: Do "Arms Races" Arise in Artificial Evolution?. Artif. Life 4, 4 (October 1998), 311-335. DOI=http://dx.doi.org/10.1162/106454698568620 and more can be found in Nolfi & Floreano, Evolutionary Robotics, 2000 Resilience, e.g. learning the own body model also under variations of the body has been investigated in J. Bongard, V. Zykov, and H. Lipson. Resilient machines through continuous self-modeling. Science, 314(5802):1118–1121, 2006. It would be great if the authors could discuss their approach/results in the context of more related work.

Reviewer 2



This is a very well-written paper and the problem / approach is very interesting. I wonder though how the authors would propose addressing this problem when the action space is discrete? In addition, Figure 5 is a bit hard to parse. Can the authors summarize some of the statistics (such as correlation) to better demonstrate what the robot is doing?

Reviewer 3



This paper proposes a generalization of GAN, which the authors refer to as Turing Learning. The idea is to let the discriminator to interact with the generator, and thus work as an active interrogator that can influence the sampling behavior of the generator. To show that this strategy is superior to existing GAN which has the discriminator to function passive responder, the authors perform two case studies of the proposed (high-level) model (interactive), comparing against models with passive discriminators. The results show that the interactive model largely outperforms the passive models, with much less variances across multiple runs. Pros 1) The paper is written well, with clear motivation. The idea of having the discriminator to work as active interrogator to influence the generator is novel and makes perfect sense. 2) The results show that the proposed addition of discriminator-generator interaction actually helps the model work more accurately. Cons 3) The proposed learning strategy is too general while the implemented models are too specific to each problem. I was hoping to see a specific GAN learning framework that implements the idea, but no concrete model is proposed and the interaction between the discriminator and the generator is implemented differently from model to model, in the two case studies. This severely limits the application of the model to new problems. 4) The two case studies consider toy problems, and it is not clear how this model will work in real-world problems. This gives me the impression that the work is still preliminary. Summary In sum, the paper presents a novel and interesting idea that can generalize and potentially improve the existing generative adversarial learning. However, the idea is not implemented into a concrete model that can generalize to general learning cases, and is only validated with two proof-of-concept case studies on toy problems. This limits the applicability of the model to new problems. Thus, despite its novelty and the potential impact, I think this is a preliminary work at its current status, and do not strongly support for its acceptance.